# Histone Deacetylase 4 Inhibition Reduces Rotenone-Induced Alpha-Synuclein Accumulation via Autophagy in SH-SY5Y Cells

**DOI:** 10.3390/brainsci13040670

**Published:** 2023-04-16

**Authors:** Luxi Wang, Ling Liu, Chao Han, Haiyang Jiang, Kai Ma, Shiyi Guo, Yun Xia, Fang Wan, Jinsha Huang, Nian Xiong, Tao Wang

**Affiliations:** 1Department of Neurology, The First Affiliated Hospital of Wenzhou Medical University, Wenzhou 325000, China; 2Department of Neurology, Union Hospital, Tongji Medical College, Huazhong University of Science and Technology, Wuhan 430022, China

**Keywords:** Parkinson’s disease, histone deacetylases, histone deacetylase inhibitor, gene therapy, neuroprotection

## Abstract

(1) Background: Parkinson’s disease (PD) is the most common movement disorder. Imbalanced protein homeostasis and α-syn aggregation are involved in PD pathogenesis. Autophagy is related to the occurrence and development of PD and can be regulated by histone deacetylases (HDACs). Various inhibitors of HDACs exert neuroprotective effects within in vitro and in vivo models of PD. HDAC4, a class Ⅱ HDAC, colocalizes with α-synuclein and ubiquitin in Lewy bodies and also accumulates in the nuclei of dopaminergic neurons in PD models. (2) Methods: In the present study, the gene expression profile of HDACs from two previously reported datasets in the GEO database was analyzed, and the RNA levels of HDAC4 in brain tissues were compared between PD patients and healthy controls. In vitro, SH-SY5Y cells transfected with HDAC4 shRNA or pretreated with mc1568 were treated with 1 μM of rotenone for 24 h. Then, the levels of α-syn, LC3, and p62 were detected using Western blot analysis and immunofluorescent staining, and cell viabilities were detected using Cell Counting Kit-8 (CCK-8). (3) Results: HDAC4 was highly expressed in PD substantia nigra and locus coeruleus. Mc1568, an inhibitor of HDAC4, decreased α-synuclein levels in rotenone-treated SH-SY5Y cells in a concentration-dependent manner and activated autophagy, which was impaired by rotenone. The knockdown of HDAC4 reversed rotenone-induced α-syn accumulation in SH-SY5Y cells and protected the neurons by enhancing autophagy. (4) Conclusions: HDAC4 is a potential therapeutic target for PD. The inhibition of HDAC4 by mc1568 or a gene block can reduce α-syn levels by regulating the autophagy process in PD. Mc1568 is a promising therapeutic agent for PD and other disorders related to α-syn accumulation.

## 1. Introduction

Parkinson’s disease (PD) is the most common movement disorder and affects about 1% of the world population over 60 years of age [1]. PD is characterized by the progressive loss of dopaminergic neurons in the substantia nigra pars compacta and the presence of cytoplasmic inclusions named Lewy bodies (LBs), in which alpha-synuclein (α-syn) is a major component of the aggregates [2,3]. There is overwhelming evidence that imbalanced protein homeostasis and α-syn aggregation are involved in familial and sporadic PD pathogenesis [3,4].

One pathway of α-syn degradation in cells is the autophagy/lysosomal pathway (ALP) [3]. Among the mechanisms possibly responsible for a-syn misfolding and aggregation and the dysfunction of the degradation pathways, the autophagy-lysosome system has attracted much attention. Impaired autophagy is inextricably related to the occurrence and development of PD [3,5]. Multiple risk genes of PD are associated with ALP, including scavenger receptor class B member 2 (SCARB2), lysosomal integral membrane protein type 2 (LIMP2), and sphingomyelin phosphodiesterase 1 (SMPD1) [6]. Moreover, direct and indirect roles have been demonstrated for numerous other PD-related mutations in the process of autophagy, such as leucine-rich repeat kinase 2 (LRRK2), MAPT, and SNCA [7,8]. A number of studies have found that autophagy levels are regulated in the substantia nigra tissues of PD patients [6]. Furthermore, the activation of autophagy may delay the disease process of PD. The overexpression of autophagy-related genes, such as factor EB (TFEB), lysosome-associated membrane protein type 2A (LAMP2A), and beclin-1, as well as autophagy inducers, such as rapamycin and lithium, can all exhibit neuroprotective effects in PD models [9,10,11,12,13,14,15,16,17].

Histone acetylation is a dynamic epigenetic process regulated by histone acetyl transferases (HATs) and histone deacetylases (HDACs) and modulates the expression of genes [18]. Previous studies have confirmed that the activity of the autophagy degradation pathway is regulated by HDACs. The acetylation of histone H3 lysine 56 (H3K56ac) and the acetylation of histone H4 lysine 16 (H4K16ac) are closely related to the regulation of the autophagy pathway [19]. HDACs may be involved in autophagy regulation by affecting the acetylation levels of these histone sites and autophagy-related ATG proteins [19]. Back in 2004, the use of HDAC inhibitors (HDACIs) linked autophagy to histone modifications for the first time. Shao and colleagues found that the HDACIs butyrate and suberoylanilide hydroxamic acid (SAHA) induced autophagic cell death in multiple human cancer cell lines [20].

For degenerative disorders, HDAIs have been intensively explored as new therapeutics [21]. In the last two decades, we have witnessed the growing field of research on epigenetic-based treatment in PD [22]. High histone acetylation levels were found to be closely related to the onset and progression of PD [23]. Various histone deacetylase inhibitors, including valproate (VPA), trichostatin A (TSA), SAHA, butyrate, and benzyl butyrate, have been proven to exert neuroprotective effects in both in vitro and in vivo models of PD [18]. However, the vast majority of these blockers are pan-HDAC inhibitors, and each pan-HDAC inhibitor blocks eight to ten HDAC subtypes simultaneously [24]. In short-term studies, the non-specific effects of these broad-spectrum inhibitors are likely to be neglected, while the consequent complex bio-effects are a potential threat to safety in their therapeutic applications for longer periods of time.

Among all 18 HDACs found in the human body, HDAC4 has a high level of expression in brain tissue [25] and also plays an important role in the development and normal function of the brain, neurodegeneration, and neuronal cell death [26]. It has been reported that HDAC4 colocalizes with α-synuclein and ubiquitin in LBs [27]. While HDAC4 is normally located in the cytoplasm, the nuclear accumulation of HDAC4 was observed in dopaminergic neurons overexpressing A53T mutant α-synuclein treated with MPP+/MPTP in vitro and in vivo, indicating that HDAC4 can be a crucial player in the pathology of PD [28]. Of note, research by Lang and colleagues revealed that HDAC4 was an upstream regulator of PD progression, and HDAC4-modulating compounds, such as tasquinimod, had great therapeutic potential for neuroprotection in PD [29].

Based on previous findings, we hypothesized that the inhibition of HDAC4 could reduce the aggregation and accumulation of α-synuclein and retard the progression of PD. Here, we attempted to selectively inhibit HDAC4 in the rotenone-induced SH-SY5Y cell model of PD, and explore its effects on α-syn levels, autophagy activity, and cell viability and, therefore, to seek scientific evidence for a new targeted therapy for PD.

## 2. Materials and Methods

### 2.1. Data Source and Data Processing

GSE43490 and GSE7621 raw data were downloaded from the Gene Expression Omnibus (GEO) (https://www.ncbi.nlm.nih.gov/geo/query/acc.cgi?acc=GSE43490, accessed on 14 January 2013; https://www.ncbi.nlm.nih.gov/geo/query/acc.cgi?acc=GSE7621, accessed on 25 April 2007). Data from GSE7621 were log base 2 transformed. The quantile normalization of the gene expression was performed using the normalize Between Arrays function in limma. Plots were generated with Graphpad Prism 8 and the ggplot2 package (ggplot2.org) in R software v3.3.3.

### 2.2. Chemicals and Reagents

Mc1568, DMSO, and Hoechst 33,342 were purchased from SIGMA-ALDRICH. DMEM/F12 medium was purchased from Hyclone. Fetal bovine serum was purchased from Four-season biological engineering Co., Ltd., Hangzhou, China. Streptomycin-penicillin solution, Cell Counting Kit-8 (CCK-8), BCA assay kit, and RIPA lysis buffer were purchased from Beyotime Biotechnology. Neofect reagent was purchased from Neofect (Beijing, China) Biotech Co., Ltd.

### 2.3. Cell Culture

The human neuroblastoma cell line (SH-SY5Y) was cultured in DMEM/F12 medium with 10% fetal bovine serum, 100 µg/mL of streptomycin, and 100 U/mL of penicillin. The cells were maintained at 37 °C in a humidified incubator with an atmosphere of 5% CO_2_. The culture medium was removed every two days and a sub-culture was used once cells reached 80–90% confluence.

### 2.4. Gene Transfection

ShRNA interference was performed by treating the cells with HDAC4 shRNA or control shRNA in 6-well plates for 24 h using a Neofect reagent as indicated in the instructions provided by the manufacturer. SH-SY5Y cells were then exposed to rotenone or vehicle for 48 h. After these treatments, cells were used for biochemical analysis.

### 2.5. Cell Viability Evaluation through CCK-8 Assay

The CCK-8 assay was performed to determine cell viability. Briefly, the SH-SY5Y cells were seeded in 24-well plates at a density of 6.7 × 10^3^ cells/well, incubated for 24 h, and then treated with reagents. This was followed by incubating with 25 µL of CCK-8 buffer in 75 µL of medium for 2 h at 37 °C, following the instructions provided by the kit company (Beyotime Biotechnology, Shanghai, China). A microplate reader (BioTek, Winooski, VT, USA) was used to measure the absorbance at 450 nm. All samples were assessed in quintuplicate.

### 2.6. Western Blot

Total proteins were extracted from the cell lysates using RIPA lysis buffer supplemented with a protease inhibitor cocktail (Sigma-Aldrich, St. Louis, MO, USA, 1:50 dilution). Protein concentrations of the extracts were measured with a BCA assay kit. For Western blot, samples were mixed with sodium dodecyl sulfate (SDS) loading buffer and separated on 8%,10%, 12%, or 15% SDS-polyacrylamide gel electrophoresis (PAGE). Proteins were electrophoretically transferred to a polyvinylidene difluoride (PVDF) membrane and blocked with 5% milk dissolved in Tris-buffered saline containing 0.1% Tween 20 (TBST) for 1 h at room temperature. The membranes were probed overnight at 4 °C with the following primary antibodies: anti-α-synuclein antibody (Abcam, Cambridge, UK, ab138501, 1:1000 dilution); anti- p62 antibody (Abcam, ab91526, 1:1000 dilution); anti-LC3B antibody (Novus, St. Charles, MO, USA, NB100-2220, 1:1000 dilution); anti-HDAC4 antibody (Proteintech, Wuhan, China, 17449-1-AP, 1:600 dilution); anti-β-actin antibody (Antgene, Wuhan, China, ANT009, 1:1000 dilution). After incubation with horseradish peroxidase-conjugated secondary antibodies (1:1000 dilution) for 1 h at room temperature, western blots were revealed through chemiluminescence. The protein levels were quantified through densitometry using ImageJ v1.47 software.

### 2.7. Immunofluorescent Staining

SH-SY5Y cells were fixed with 4% paraformaldehyde for 10 min, washed in phosphate-buffered saline (PBS), and permeabilized with 0.5% Triton X-100 in PBS for 20 min at room temperature. The cells were saturated with 1% bovine serum albumin in PBS for 30 min at room temperature and processed for immunofluorescence with anti-α-synuclein antibody and anti-LC3 antibody followed by Cy3-conjugated anti-rabbit IgG. Nuclear morphology was analyzed after staining with the fluorescent dye Hoechst 33,342 for 10 min. Images from the sections were examined on an OLYMPUS IX71 fluorescent microscope (OLYMPUS, Tokyo, Japan).

### 2.8. Quantitative Real-Time PCR (qRT-PCR)

Total RNA was isolated from SH-SY5Y cells using RNAiso Plus (TaKaRa, Gunma, Japan). Total RNA (2 μg) was reverse transcribed to cDNA using the PrimeScript^TM^ Ⅱ 1st Strand cDNA Synthesis Kit (TaKaRa, Japan) to determine the mRNA expressions of α-syn through qRT-PCR using the SYBR Green reagent (TaKaRa, Japan). The PCR condition was as follows: 95 °C for 5 min, 60 °C for 20 s, and 40 amplification cycles. Housekeeping gene b-actin served as an internal control. Data analysis is based on the ΔΔCt method with the normalization of raw data to β-actin. Each reaction was run in triplicate. a-Syn primer: forward, forward: 5′-AGGAATTCATTAGCCATGGATGTATTC -3′; reverse: 5′-AGATATTTCTTAGGCTTCAGGTTCGTAGT-3′; β-actin primer: forward: 5′-GACCCAGATCATGTTTGAGA-3′; reverse: 5′-GCTTGCTGATCCACATCTGC-3′.

### 2.9. Statistical Analysis

All values are expressed as mean ± SEM of three independent experiments carried out in triplicate. Statistical tests were performed using Student’s t-test or a one-way ANOVA with a least significant difference (LSD) post-hoc test. All statistics were calculated using SPSS 20.0 software, and a value of *p* < 0.05 was considered statistically significant.

## 3. Results

### 3.1. HDAC4 Was Highly Expressed in PD Brain Tissues

Two previously reported datasets in the GEO database (GSE43490 and GSE7621) were selected to verify whether HDAC4 was differentially expressed in PD. GSE43490 was an RNA-Seq dataset of post-mortem explants from the dorsal nucleus of the vagus nerve, locus coeruleus, and substantia nigra obtained from controls and Braak 4 (BK4) and 5 (BK5) stages of PD patients. GSE7621 was an RNA-Seq dataset of substantia nigra from postmortem brains of patients with PD. The HDAC4 gene site selected from GSE43490 was A_23_P210048, and those from GSE7621 were 204225_at and 228813_at. The heatmap revealed that HDAC4 seemed to express highly in the brain tissues of PD patients (Figure 1A). The statistical analysis of GESE43490 showed that HDAC4 was significantly upregulated in PD locus coeruleus and tended to be increased in PD substantia nigra, with no statistically significant differences (Figure 1B,C). Our results from GSE7621 showed that HDAC4 was significantly increased in the substantia nigra of PD patients (Figure 1D,E).

### 3.2. Mc1568 Decreased α-syn Levels in SH-SY5Y Cells in a Concentration-Dependent Manner

In this study, mc1568 was used to selectively inhibit the activity of HDAC4. The SH-SY5Y cells were treated in the presence of both mc1568 and rotenone. Western blot showed that α-syn levels in rotenone-treated neurons were decreased by mc1568 in a concentration-dependent manner (Figure 2A,B). We also detected the effect on the cell viability of mc1568 using CCK-8. Mc1568 (0.01 μM to 10 μM) did not significantly affect the viability of rotenone-treated SH-SY5Y cells (Figure 2C). These results demonstrated that mc1568 (0.1 μM to 10 μM) significantly reversed the abnormal accumulation of monomeric α-syn induced by rotenone without remarkable cytotoxic effects in SH-SY5Y cells.

### 3.3. Mc1568 Activated Autophagy in SH-SY5Y Cells

To investigate in which way mc1568 reduced α-syn levels in SH-SY5Y cells, the levels of autophagy-related proteins, LC3-Ⅰ and LC3-Ⅱ, were examined through Western blot. Our results showed that exposure to rotenone decreased the LC3-Ⅱ/LC3-Ⅰ ratio while mc1568 increased the ratio significantly, thus indicating that mc1568 activated autophagy, which was impaired by the treatment of rotenone in SH-SY5Y cells (Figure 3).

### 3.4. Knockdown of HDAC4 Reversed Rotenone-Induced α-syn Increase in SH-SY5Y Cells

To confirm the neuroprotective effects of HDAC4 downregulation, we knocked down the expression levels of HDAC4 in SH-SY5Y cells (Figure 4) and detected the levels of α-syn proteins. Western blot analysis revealed that monomeric α-syn levels in SH-SY5Y cells receiving the treatment of rotenone were significantly increased (Figure 5A,B), while the knockdown of HDAC4 abolished such a pathological accumulation of α-syn induced by rotenone (Figure 5A,B). These results were confirmed through immunofluorescent staining (Figure 5C). Moreover, the mRNA levels of α-syn were determined through qRT-PCR, which revealed that the knockdown of HDAC4 had no significant effect on α-syn gene transcription (Figure 5D). Rotenone induced the aggregation of α-syn in SH-SY5Y cells, which was rescued by the knockdown of HDAC4 (Figure 5E).

### 3.5. Knockdown of HDAC4 Protected SH-SY5Y Cells against Rotenone-Induced Autophagy Impairment

To determine whether the autophagy pathway is involved in the neuroprotective effects of HDAC4 downregulation, the levels of autophagy-related proteins were examined through Western blot, and autophagosome formation was observed via LC3B fluorescent puncta using immunofluorescence. Our results showed that exposure to rotenone reduced the LC3-Ⅱ/LC3-Ⅰ ratio and increased p62 levels significantly (Figure 6A,B,D), thus indicating that rotenone induced autophagy impairment in SH-SY5Y cells. Meanwhile, the knockdown of HDAC4 in rotenone-treated SH-SY5Y cells led to an increase in the LC3-Ⅱ/LC3-Ⅰ ratio and a decrease in p62 levels, indicating a higher level of autophagy activity (Figure 6A,B,D). Consistently, the formation of LC3B fluorescent puncta representing autophagosomes was markedly reduced with the treatment of rotenone but was increased by the knockdown of HDAC4 (Figure 6C). These results implied that HDAC4 downregulation promoted the clearance of abnormally accumulating α-syn via activating the autophagy pathway, which was impaired in rotenone-treated SH-SY5Y cells.

## 4. Discussion

The neuroprotection effects of HDAC inhibitors have been broadly demonstrated [18]. These HDACIs include VPA, butyrate, and phenylbutyrate, which inhibit class Ⅰ HDACs (HDAC1, 2, 3, and 8) and class Ⅱa HDACs (HDAC4, 5, 7, and 9), as well as TSA and SAHA, which inhibit class Ⅰ, class Ⅱa, and class Ⅱb HDACs (HDAC6 and 10) [18]. Chronic VPA treatment protected the dopaminergic neurons in the substantia nigra and the dopaminergic terminals in the striatum in 6-OHDA hemi-parkinsonian rat model [30]. Phenylbutyrate dose-dependently decreased the degeneration of nigral dopaminergic neurons and avoided the development of motor deficits in a rotenone-induced mouse model of PD [31].

As one of class Ⅱa HDACs, HDAC4 is expressed throughout the body with enrichment in the brain and relates closely to brain development, neuronal cell death, memory impairment, and neurodegeneration [25,26,32]. Previous findings have demonstrated that HDAC4 may be involved in the pathology of PD. HDAC4 colocalized with α-syn and ubiquitin in LBs, the pathological hallmarks of PD, and was more likely to be in the core or peri-core of LBs [27]. In our study, we found that HDAC4 was highly expressed in the substantia nigra and dorsal nucleus of the vagus nerve and the locus coeruleus of PD patients. Moreover, we found that the inhibition of HDAC4 ameliorated the increase in α-syn levels and protected against autophagic deficits in the rotenone-induced cell model, thus implying its neuroprotective action in PD.

Mc1568 is a selective inhibitor of HDAC4, 5, and 6 with tissue-specific activity [33]. Past studies have demonstrated that mc1568 induced the downregulation of HDAC 4 and HDAC5 in neurons, while it did not block HDAC6 in both in vitro cultured neurons and in the brain in vivo [33,34,35]. Mc1568 has previously been reported to mediate neuronal protection. It could promote neurite growth in dopaminergic and sympathetic neurons and protect neurons against MPP+-induced neurotoxicity [36]. Moreover, motor activity changes in rats, induced by thimerosal (a neurotoxic organic compound), could be reduced by mc1568 [34]. In our study, mc1568 treatment reversed the abnormal accumulation of α-syn in a dose-dependent way. We speculated that mc1568 reduced α-syn monomers by inhibiting HDAC4. Our data revealed that repressed expression of HDAC4 also brought about a decrease in α-syn levels in rotenone-treated SH-SY5Y cells, further confirming the neuroprotective effects of HDAC4 downregulation. Two neuroprotection studies have been conducted using other inhibitors of HDAC4. A small molecule inhibitor of HDAC4 and 5, LMK235, was found to protect against MPP^+^- and α-syn-induced loss of dopaminergic neurons in cellular models of PD [37]. In iPSC-derived dopamine neurons from PD patients, Lang and colleagues demonstrated that tasquinimod, an allosteric inhibitor of HDAC4, could alleviate ER stress, rescue defective ALP and reduce α-syn release [29]. These results demonstrated the potential of HDAC4 as a therapeutic target for PD. Notably, it was shown that HDAC4 presented pathological nuclear accumulation in PD, although it was located in the cytoplasm under normal physiological conditions [28]. Interestingly, past studies have demonstrated that mc1568 was able to facilitate the degradation of HDAC4 and HDAC5 in neuronal cell nuclei, further indicating that mc1568 might be a promising strategy for delaying PD progression [34,38].

To prevent side effects resulting from epigenetic therapy, it is important for drug treatment to gain large neuroprotective effects, even in small amounts of the drug. In our study, the remarkable effect of α-syn clearance derived from mc1568 was obtained at 0.1 μM. Naldi and colleagues reported that mc1568 did not alter the distribution of cells in the cell cycle and did not induce cell death, even at 50 μM [39]. It was also reported that mc1568 failed to inhibit cell proliferation at 5 µM, unlike other traditional pan-HDACIs [33]. Moreover, it was demonstrated that the systemic administration of mc1568 was well-tolerated by mice and rats at doses of up to 6.5 mg/kg and 40 mg/kg, respectively [34,40]. These findings imply that mc1568 holds the promise of being a safe, well-tolerated therapeutic agent for PD and other disorders related to α-syn accumulation.

In PD, native unfolded α-syn misfolds and self-aggregates to form oligomers and fibrillar structures that accumulate into LBs [41]. Yet, it is elusive which conformation of α-syn is its native state in cells. While some researchers reported that α-syn predominantly exists as an unfolded monomer, other studies found that α-syn is a folded tetramer in human neuronal and non-neuronal cell lines [3,42,43]. Whereas tetramers show little or no amyloid-like aggregation, monomers are prone to aggregation and are considered a source of a-synuclein toxicity [42,43,44]. PD-linked α-syn mutations, including A53T and E46K, decrease its tetrameric and increase its monomeric conformation, suggesting that the unfolded monomer might lead to a-synuclein toxicity [44]. Duplication or triplication of the α-syn gene was also found to cause early-onset PD, which also indicates that elevated levels of wild-type α-syn can be neurotoxic [45,46]. Consistent with previous studies, we found levels of α-syn monomers were significantly increased by rotenone in SH-SY5Y cells. Although which species of a-syn are toxic is controversial, amyloid-like insoluble fibrils observed in LB or soluble, prefibrillar intermediates, such as oligomers or protofibrils, the aggregation of α-syn underlies the pathogenesis of PD [3]. In our rotenone-induced cell model of PD, the aggregation of α-syn was increased, while the knockdown of HDAC4 resulted in a reduction in α-syn aggregates. This provides corroborating evidence for the neuroprotective role of HDAC4 modulation in α-synucleinopathies.

Overexpression, misfolding, or decreased degradation are all pathways that may contribute to the accumulation of α-syn in neurons and lead to α-syn-mediated toxicity [45,46,47]. α-Syn is degraded by two pathways, ALP and the ubiquitin-proteasome system (UPS) [3]. Previous findings have demonstrated that dysfunction in the autophagy pathway is common in numerous neurodegenerative diseases and relates closely with the accumulation of α-syn as well as the progression of PD [48,49,50]. Meanwhile, pathological α-syn can also impair autophagosome function and amplify its toxicity in neurons [41]. Here, our data showed the repression of HDAC4, whether by mc1568 or gene block, resulted in a decrease in α-syn levels with alleviated impairment of autophagy in the cell model of PD. Moreover, no significant change in the transcription levels of the α-syn gene was noted when HDAC4 was knocked down in rotenone-treated cells, indicating that the reduced protein levels of α-syn may result from elevated degradation. As such, our results implicate that HDAC4 downregulation may correct the abnormal accumulation of α-syn via the upregulation of autophagy activity.

Our data suggest that autophagy activity is reduced in rotenone-treated SH-SY5Y cells. However, previous research indicated that alterations in autophagy are inconsistent in PD in different studies [6,51]. For example, it has been found that Beclin-1 expression levels increased significantly in the substantia nigra [52], but decreased in the anterior cingulate cortex of PD patients [53]. Interestingly, the study by Lang and colleagues found neuronal autophagosome numbers, measured through LC3-Ⅱ levels, were higher in PD than in controls, which is contrary to our results. However, Lang et. also found lysosomal accumulation, measured through LAMP1, in PD patients was increased compared to controls, which means the autophagic flux was hindered in PD. Moreover, the supplementary data of their study also indicated that autophagic flux, quantified by dividing the level of LC3-Ⅱ with bafilomycin treatment by the level of LC3-Ⅱ without treatment, was lower than controls in PD models. Consistently, our data show p62 levels were significantly increased in rotenone-treated cells, which means autophagic flux was blocked. These findings are also in accordance with the previously published studies. In nigral dopaminergic neurons from PD patients, autophagosomes and lysosomal marker proteins are increased and decreased, respectively, indicating that the autophagic flux is profoundly blocked in these patients [54,55]. On the other hand, the outcomes described by Lang et al. and our results consistently demonstrated that modulation of HDAC4 corrected the perturbations in autophagic flux. The difference in results between the two studies is that Lang et al. found that the LC3-Ⅱ levels of PD models were lower than controls, while we found that they were higher. The differences in LC3-Ⅱ levels in these two studies may result from different observation time points. According to previous research, while 24 h of treatment with rotenone elevated LC3-Ⅱ levels in neurons, 7-day treatment with rotenone resulted in lower expression of LC3-Ⅱ [56]. Further, different cell model systems of PD may be another reason for the contradiction. We infer that blocking of HDAC4 may rescue disrupted autophagic flux in PD no matter whether the autophagosome number is increased or decreased, interfere with α-syn aggregation, and, therefore, halt the progression of α-syn pathology.

This study is not without limitations. We acknowledge the quantitative analysis in the present study is limited to the protein levels of monomers, but has not been applied to investigate aggregated species of α-syn (oligomers, protofibrils, and fibrils). Thus, more studies are needed to understand the effects of HDAC4 knockdown and mc1568 on α-syn aggregation. A second limitation of this study is that HDAC4 modulation has not been directly proven to be involved in the neuroprotective effects of mc1568 against rotenone-induced toxicity. Future research is required to confirm this. In addition, considering autophagic activity may be alterative at different time points of rotenone treatment, as mentioned above; further time-course studies are warranted.

## 5. Conclusions

In this study, we have shown that HDAC4 was highly expressed in PD substantia nigra and locus coeruleus. Moreover, we have demonstrated the inhibition of HDAC4, either by mc1568 or by gene block, reduced α-syn levels via the correction of autophagy defects in an in vitro model of rotenone-induced PD. These data confirm HDAC4 as a potential therapeutic target for PD. Mc1568, the selective inhibitor of HDAC4, is promising for epigenetic therapy of PD and other disorders related to abnormal accumulation of α-syn.

## Figures and Tables

**Figure 1 brainsci-13-00670-f001:**
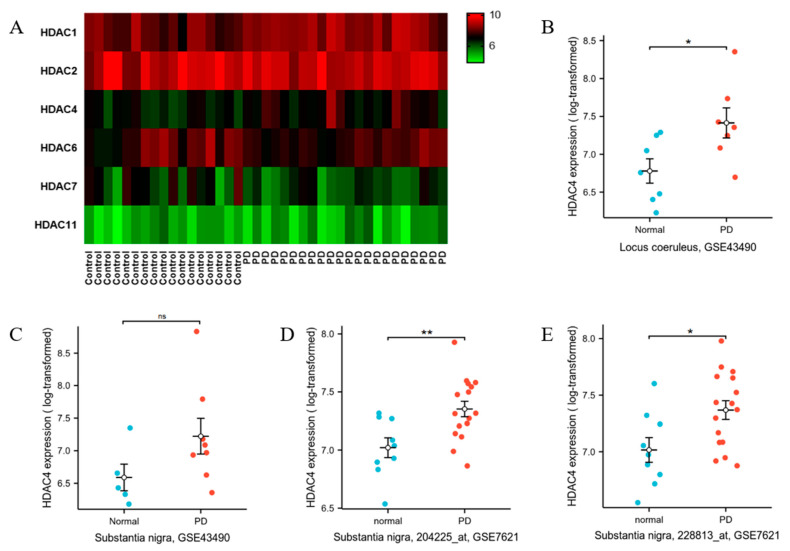
HDAC4 was highly expressed in PD brain tissues. (**A**) Gene expression profile of HDACs in PD dataset GSE43490. (**B**,**C**) Log-transformed RNA levels of HDAC4 from GSE43490 were analyzed. HDAC4 was significantly upregulated in the locus coeruleus of PD patients and tended to be increased in PD substantia nigra, with no statistically significant differences. (**D**,**E**) Log-transformed RNA levels of HDAC4 (two gene sites) from GSE7621 were analyzed. HDAC4 was significantly increased in the substantia nigra of PD patients. * *p* < 0.05; ** *p* < 0.01. ns., non-significant.

**Figure 2 brainsci-13-00670-f002:**
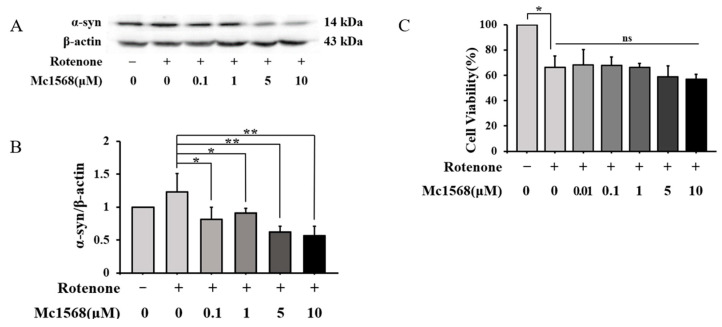
Mc1568 induced a decrease in α-syn levels in rotenone-treated SH-SY5Y cells without impairment of cell viabilities. SH-SY5Y cells were treated with indicated concentrations of mc1568 for 2 h. Then, 1 μM of rotenone or vehicle was added to the medium for another 24 h. α-Syn levels were detected through Western blot analysis. Cell viability was detected through CCK-8. (**A**,**B**) Mc1568 decreased α-syn levels in SH-SY5Y cells with the treatment of rotenone in a concentration-dependent manner. (**C**) Mc1568 had no significant influence on rotenone-treated SH-SY5Y cells. * *p* < 0.05; ** *p* < 0.01; ns., non-significant.

**Figure 3 brainsci-13-00670-f003:**
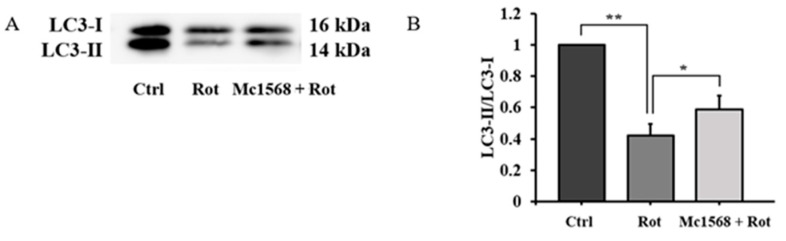
Mc1568 activated autophagy in SH-SY5Y cells. Levels of LC3 were detected through Western blot analysis. (**A**,**B**) Rotenone reduced the LC3-Ⅱ/LC3-Ⅰ ratio in SH-SY5Y cells, while mc1568 induced an increase in the LC3-Ⅱ/LC3-Ⅰ ratio. * *p* < 0.05; ** *p* < 0.01. Ctrl., control. Rot., rotenone.

**Figure 4 brainsci-13-00670-f004:**
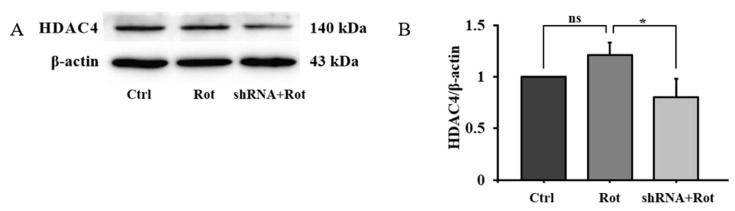
HDAC4 was knocked down in SH-SY5Y cells. SH-SY5Y cells transfected with HDAC4 shRNA or control shRNA were exposed to 1 μM of rotenone or vehicle for 48 h. HDAC4 levels were detected through Western blot analysis. (**A**,**B**) HDAC4 shRNA significantly reduced depression levels of HDAC4 in SH-SY5Y cells. * *p* < 0.05. ns., non-significant. Ctrl., control. Rot., rotenone.

**Figure 5 brainsci-13-00670-f005:**
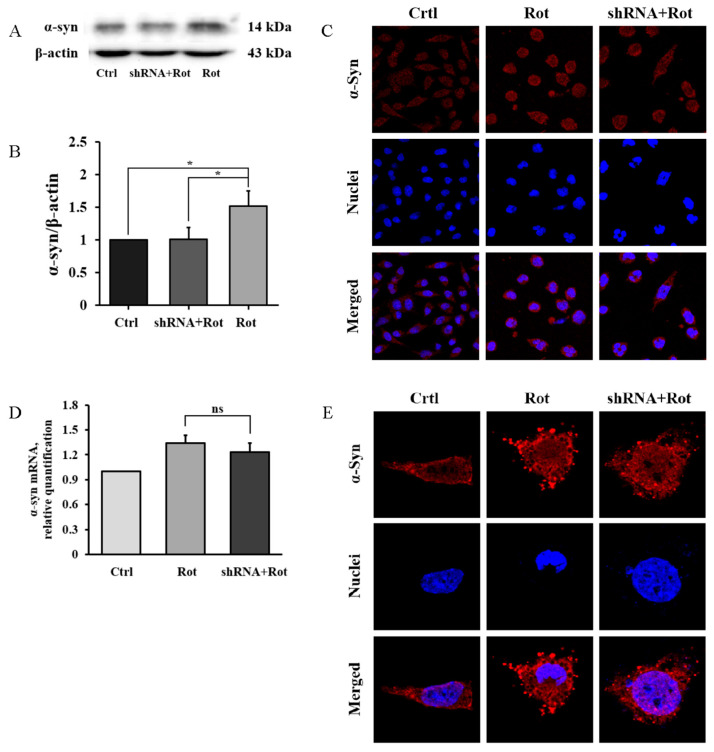
Knockdown of HDAC4 reversed the rotenone-induced increase in α-syn levels in SH-SY5Y cells. SH-SY5Y cells transfected with HDAC4 shRNA or control shRNA were exposed to 1 μM of rotenone or vehicle for 48 h. α-Syn levels were detected through Western blot analysis and immunofluorescent staining. mRNA levels of α-syn were determined through qRT-PCR. (**A**–**C**) Rotenone increased monomeric α-syn levels in SH-SY5Y cells, and the knockdown of HDAC4 reversed the increase in α-syn levels. (**D**) Knockdown of HDAC4 had no significant effect on α-syn gene transcription. (**E**) Rotenone induced aggregation of α-syn in SH-SY5Y cells, which was rescued through the knockdown of HDAC4. * *p* < 0.05. Ctrl., control. Rot., rotenone. ns., non-significant.

**Figure 6 brainsci-13-00670-f006:**
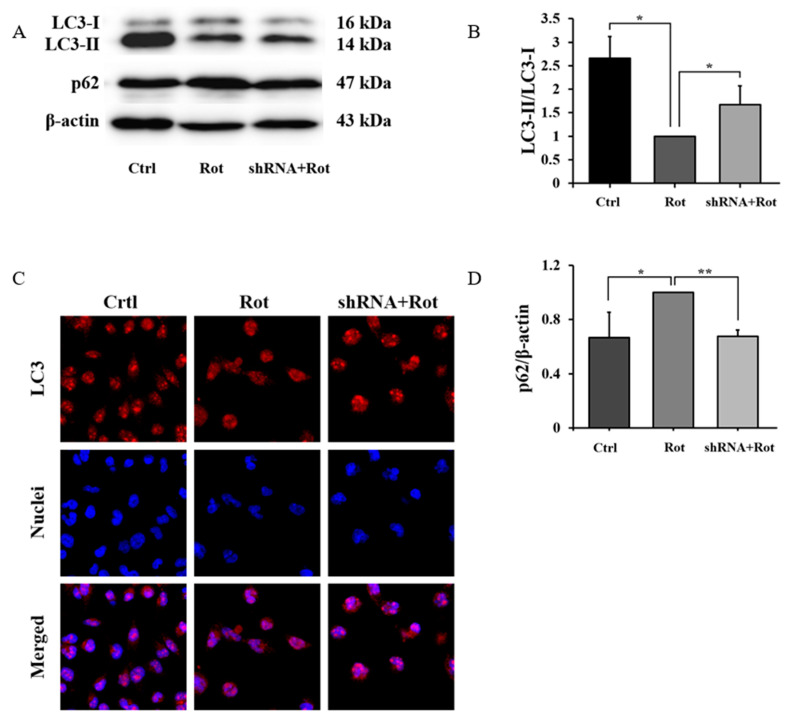
The knockdown of HDAC4 activated autophagy in SH-SY5Y cells. Levels of LC3 and p62 were detected through Western blot analysis. LC3B puncta within autophagosomes were identified using immunofluorescent staining. (**A**,**B**,**D**) Rotenone reduced the LC3-Ⅱ/LC3-Ⅰ ratio and increased p62 levels in SH-SY5Y cells, while HDAC4 downregulation induced an increase in LC3-Ⅱ/LC3-Ⅰ ratio and a decrease in p62 levels. (**C**) The knockdown of HDAC4 increased LC3B red fluorescent puncta, which were reduced with the treatment of rotenone in SH-SY5Y cells. * *p* < 0.05; ** *p* < 0.01. Ctrl., control. Rot., rotenone.

## Data Availability

Data available upon request.

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
