# Peer review of "Histone Deacetylase 4 Inhibition Reduces Rotenone-Induced Alpha-Synuclein Accumulation via Autophagy in SH-SY5Y Cells"

_brainsci, 2023, doi:10.3390/brainsci13040670_

Round 1

Reviewer 1 Report

brainsci-2262082: HDAC4 Inhibition Reduces Rotenone-Induced Alpha-Synuclein Accumulation via Autophagy in SH-SY5Y Cells

This is an excellent paper, studying the molecular mechanisms of alpha-synuclein pathology, using the HDAC4 inhibition. In the culture-cell model of Parkinson's disease, it is clearly shown that HDAC4 is highly involved in synucleinopathies. The comments are suggested as below, which may improve the value of the present study.

<Major Points>

(1) Aggregation of alpha-synuclein

The major neurotoxicity of alpha-synuclein is thought to derive from its aggregation. The present study examined only the amount of alpha-synuclein protein. Its aggregation should be examined or at least discussed.

(2) The change in mRNA of the alpha-synuclein gene by the treatment.

The amount of alpha-synuclein is probably determined from not only the protein degradation but from the change of mRNA. Because the major function of HDAC is the regulation of gene expression. In some experiments the change of mRNA should be shown to understand the molecular relationship between HDAC4 and alpha-synuclein

(3) Specificity of m1568

In the introduction Line 51-55, the specificity of HDACs inhibition is described. The HDAC inhibitor m1568 used in the present study is a so-called class 2 HDAC inhibitor which inhibits HDAC4 and some other HDACs. There are some other classes of HDAC inhibitors. The effect of these inhibitors should be examined or discussed. The knock-down of HDAC4 does not confirm that only HDAC4 is involved in synucleinopathies.

<Minor Points>

(a) The inhibition of the activity of HDACs in the present condition, using SH-SY5Y cells and the same concentration of m1568, should be shown.

(b) In Fig 5 panel C, the cell morphology by phase contrast microscopy without staining should be shown to evaluated morphological changes of the cells after the treatments.

(c) Abstract, line 14, "In this study, we reported HDAC4 was highly expressed in PD brain tissues." The most important result is not the high expression of HDAC4 but the acceleration of alpha-synuclein accumulation by HDAC4. This part should be revised.

(d) In the "2. Materials and Methods", the section of "Materials" is necessary. Especially the source of mc1568 should be clarified.

(e) Usually, the manuscript PDF of MDPI for the review is in the actual publication format. However, the present PDF is that of the classical manuscript, in which the figures are separately assigned after the text. When this report is accepted, the format should be reassigned.

End of File

Reviewer 2 Report

Review of a manuscript “HDAC4 Inhibition Reduces Rotenone-Induced Alpha-Synuclein Accumulation via Autophagy in SH-SY5Y Cells 3” by Luxi Wang and coauthors.

Parkinson’s disease is the second after Alzheimer’s disease neurodegenerative disorder, for which there is no efficient treatment modifying its course.  A hallmark of this diseases is accumulation of toxic aggregates of alpha-synuclein leading to neuron’s death. The authors investigated a role of histone deacetylase in the pathogenesis of Parkinson’s disease. This is an important field of biomedical research and the results presented in the manuscript will be interesting for the readership of the journal.

The following corrections should be made.

Abstract

“SH-SY5Y cells transfected with HDAC4 shRNA…”  “HDAC4 was highly expressed in PD brain tissue” These two sentences are confusing. In the first the authors state that they used a cell model, in the second they mention brain tissue. These discrepancies should be eliminated.  

Introduction

“In the meanwhile, the last two decades have witnessed the growing field of research on epigenetic 45

treatment in PD. This sentence should be corrected and the reference should be added as follows:” In the last two decades we have witnessed the growing field of research on epigenetic-based  treatment in PD [reference: ” alpha-Synuclein and mechanisms of epigenetic regulation. Brain Sciences, 2023, 13, 150. https://doi.org/10.3390/.]

 Materials and Methods

“This was followed by incubating with 25µl CCK-8 buffer in 75µl medium for 2 hours at 37 following the instructions provided by the kit company.” The authors should give the name of the company.

 “Total proteins were extracted from the cell lysates using RIPA lysis buffer (Beyotime Biotechnology) supplemented with peotease inhibitor cocktail (Sigma-Aldrich).. The concentration of the buffer and inhibitor cocktail should be given. “Peotease” should be corrected as “protease” .

 “The membranes were probed overnight at 4 with the following primary antibodies” Dilution of both primary and secondary antibodies should be given.

Results

“In this study, mc1568 was used to selectively inhibit the activity of HDAC4 [25].” The authors write “In this study”, but give the reference on Nebbioso, A., et al.  If they used the method described by Nebbioso, A., et al, they should state it clearly.

Figure 4. There is discrepancy between the intensity of the HDAC4 bands on A and the results of their scanning on B. On A the band corresponding to Tot looks smaller then the band corresponding to control, whereas on scanning the peak for Rot looks higher than for  control. This should be explained.

Figure 6. The order of samples on A is Ctrl – Rot – shRNA+Rot. On B it looks different: Ctrl -  shRNA+Rot – Rot. It is unclear why the authors changed the order of samples between A and B, since it makes more complicated the understanding of the results.

Discussion

“Our data suggests that autophagy activity is reduced in rotenone-treated SH-SY5Y cells. However, previous researches indicated that alterations in autophagy are inconsistent in PD in different researches [8, 33].” The authors should explain, at least hypothetically, the reason of the contradiction of their results compared to the literature data.

“researches” should be written as research.

Reviewer 3 Report

The manuscript is an extremely preliminary set of data indicating a potential role of HDAC4. The material are poorly explored and add little to existing literature. Significant additional experiments to characterize and prove these phenomena are required. The authors cannot say this provides a novel avenue given the paper from Wade Martin's lab in 2019, Cell Stem Cell. The paper entirely misses the important contact that non-specific HDAC inhibitors are activators of autophagy. 

The literature review is completely inadequate, several papers previously describing the effect of MC1568 are ignored despite being in PD models. For instance; Collins et. al 2014 https://link.springer.com/article/10.1007/s12035-014-8820-8 

There are no obvious indication of the number of replicates. 

It is not obvious that there is an effective knockdown of HDAC4. Needs to be repeated and generate a better KD. 

Figure 6c - There is no difference that is obvious, needs quantification. 

Figure 2: Shows no increase in Alpha synuclein under rotenone, I appreciate the conditions are different, but this strongly indicates the need for time courses. 

All of the experiments need time courses, because rotenone exposure activates autophagy at various time points and concentrations independent of the authors treatment. 

The western blots are not convincing, full blots should be provided, the difference observed does not fit the quantification data, please provide individual replicate data. 

The statistical analysis is not scientifically valid. For each figure with more than two groups, an adjustment needs to be made for multiple comparisons. For the data in figure one there is no adjustment for genome wide comparisons. It is not clear what the array data is, - is this log expression, the axis should be more clearly labelled. 

Figure one is not scientifically correct. The groups have all been combined but these are different brain regions. When separately analyzed there is no statistically significant downregulation. Please justify. (after adjustment)

Other labs data is incorrectly described. In our study should read, analysis of previously published datasets!

The microscopy needs scale bars, closer imaging, and better quality. 

The authors need to fully engage with the work by the Wade Martins lab in both discussion and introduction published in Cell stem cell over 4 years ago. 

Round 2

Reviewer 3 Report

The authors did not generate a point by point response to all criticisms. This needs to be done. Please review this document and respond to every point, not selectively. 

The following experiments as part of the major revision need to be completed. 

It is not obvious that there is an effective knockdown of HDAC4. Needs to be repeated and generate a better KD. 

After reviewing the original blot provided by the authors. It is clear this experiment did not work. The authors need to generate new data or remove.

All of the experiments need time courses, because rotenone exposure activates autophagy at various time points and concentrations independent of the authors treatment. 

- Because there are variations in this response and indeed that of alpha synuclein, we need time courses to determine if these effects are robust or transitory. Because the time affects these phenotypes so markedly this is an essential experiment. 

The authors need to correct each experimental analysis for multiple comparisons. 

The statistical analysis has not been corrected for figure 1. The normalization procedure is not the correct way to respond to the criticism. The statistical analysis needs to be corrected for multiple comparisons at a genome level to answer the question if this is an authentically differentially expressed gene. For instance through a benjamini hochberg correction or other correction for the FDR.

https://biostats.bepress.com/ucbbiostat/paper110/

Minor; 

Please correct all the reference, many have an error source not found in this version. 
